# Putting the Boom, Boom, Boom into Physical Activity and Health: Music Festivals as a Positive Health Alternative to Couch Fandom

**DOI:** 10.3390/ijerph16122105

**Published:** 2019-06-14

**Authors:** Eddie Bradley, Lauren Close, Ian Whyte

**Affiliations:** 1Department of Sport & Exercise, University of Sunderland, Sunderland SR1 3SD, UK; ian.whyte@sunderland.ac.uk; 2Students Union, Teesside University, Middlesbrough TS1 3BA, UK; lauren.close@hotmail.co.uk

**Keywords:** festivals, fitness, physical activity guidelines, movement intensities, GIS

## Abstract

*Background*: Despite the popularity of outdoor music festivals in the UK, no evidence exists of the volume or intensity of movement that occurs through attendance at these festivals and the potential health benefits this may provide. The aim of this study was to accurately record the amount of physical activity and movement at the Glastonbury Festival and to compare it against recommended levels. *Methods*: 22 attendees wore an Actigraph activity monitor and GPS data-logger to the Glastonbury Festival. Distances travelled, speeds and durations were recorded. Activity levels were identified based on step count thresholds and the total duration spent in moderate to vigorous physical activity (MVPA) was calculated. *Results*: Mean total distance of 66.1 km was recorded with daily distance (11.01 km), movement duration (11 h 28 min) and steps/day (15,661). Total MVPA of 927 min occurred over the festival period. *Conclusions*: This study objectively recorded the volume of physical activity that occurred at an outdoor UK festival. Large movement distances and MVPA six times greater than the recommended guidelines for health benefits were found. It can be suggested that attendance at large-scale festivals can be used as a modality for attaining physical activity guidelines alongside commonly suggested fitness activities.

## 1. Introduction

Physical inactivity has been identified in the UK as a major health issue that is associated with lower quality of life [1] and an increased incidence of chronic disease such as coronary heart disease [2] and diabetes [3], alongside increased levels of obesity [4]. The recommended level of physical activity (PA) in the UK is 150 min of moderate to high intensity exercise per week [5], with 66% of men and 58% of females achieving this total [6]. The recent Active Lives Report [7] reported that 25.2% of UK adults completed zero sessions of “30 min of at least moderate intensity activity in the previous 28 days” and such levels are a concern.

Behavioural interventions aimed at increasing PA levels have been shown to have beneficial effects on a range of health issues [8]. Such interventions include mass media or internet promotion, large population-based campaigns and targeted programmes. Cobiac et al. [9] identified that activity programmes centred on the use of pedometers in Australia, while being supported by mass media promotion, was the most cost-effective method. Similarly, Pillay et al. [10] reported that a 10-week pedometer-based intervention in South Africa produced PA improvements of nearly 1000 steps/day per person and was perceived as beneficial by participants. Physical activity interventions that increase walking as a mode of activity are effective tools in PA promotion, but the efficiency of these programmes is related to their specific intervention approach. Monitors have been used to measure PA in adults and children in large-scale surveys, such as the National Health and Nutrition Examination Survey (NHANES) in the USA [11] and the Health Survey for England [12], and have been validated in previous studies [13,14]. These devices usually consist of an accelerometer that determines the orientation of the device and whether movement is occurring. The combination of these two characteristics enables the activity monitor to identify vertical oscillations of the whole-body centre-of-mass as a result of walking and measure step count. In-built device algorithms take the step count as counts of movement and calculate energy expenditure in terms of metabolic equivalents (METS) determined previously through exercise testing using indirect calorimetry [15,16]. Vector magnitude (VM) cut points are then utilised to define PA intensities, with high accuracies reported [17]. It is also possible to use global positioning systems (GPS) to map movement patterns during PA and accurately record movement using satellites to locate a device and create spatial and temporal markers. Kang et al. [18] used data from 706 adults to classify accelerometer derived PA bouts to define walking behaviour. Similarly, Krenn et al. [19] reviewed the existing literature on the relationship between GPS data and PA. From 24 studies they found evidence that GPS is a useful tool in defining the location of PA bouts, that the strength of these GPS studies is not affected by sample size or data collection length, but limitations exist relating to data loss and participant adherence that must be overcome. The combination of both activity monitors and GPS create powerful tools [20,21] that can be applied in an integrated manner to assess PA behaviour in a range of challenging environments.

Over the past 20 years there has been a boom in the UK music festival scene [22,23] with 3.9 million people attending festivals in the UK in 2017 [24]. The outdoor nature of these events coupled with site layouts that necessitate movement between stages and campsites create a potential environment for promoting PA. However, currently there is no evidence of the volume or intensity of movement that occurs through attendance at these festivals. Only anecdotal data (for example https://www.gigwise.com/news/101497/miles-walked-and-calories-burned-at-festivals-glastonbury-v-festival; https://www.efestivals.co.uk/forums/topic/213951-how-far-did-you-walk/) exist from people who have attempted casual use of pedometers or mobile phone devices that record for only a portion of the festival. Therefore, the aim of this study was to accurately record the movement patterns and amount of physical activity at the Glastonbury Festival of Contemporary Performing Arts and to explore the extent to which attendance at a festival has the potential to be beneficial to health.

## 2. Materials and Methods

A convenience sample of fourteen males and eight females, recruited through word-of-mouth, agreed to participate in the study. Characteristics are outlined in Table 1. Pre-participation physical activity levels were obtained through the self-report Par-Q questionnaire, and this ranged from low (n = 5; e.g., asthma suffer) to medium (n = 15; e.g., weekly 5-a-side player) to high (n = 2; e.g., 18 mile/week runner). All participants were regular festival attendees, being known to the authors (themselves attendees), having attended the Glastonbury Festival multiple times along with a range of other smaller festivals over the past 15 years. This facilitated participant recruitment and increased the potential for study adherence. Ethical approval (UoS 181) was obtained from University of Sunderland Ethics Committee and the Declaration of Helsinki was adhered to throughout.

Glastonbury Festival is the largest greenfield festival in the world. It is a six-day (Wednesday 8:00–Monday 17:00) long event held on 900 acres (3.64 km^2^) of Somerset farmland, centred on Worthy Farm, Pilton [25]. The site is approximately 1.8 km east–west and 2.3 km north–south with a perimeter fence of 7.7 km in length, with similar sized car-parks to the east and west of the site. The current licence allows for up to 177,500 people to be on the site at any one time with 135,000 tickets sold to the general public. The general topography of the site is a valley running east–west across the site sloping down towards the centre of the site with steep northern and southern edges.

Physical activity levels were recorded using Actigraph GT3X+ triaxial accelerometer activity monitors (Actigraph Corp., Pensacola, FL, USA) attached to participants’ waistbands and positioned on left hips. Epoch values were set to 10 s and recorded continuously from Wednesday 8:00 until Monday 14:00, with the devices worn at all times including during sleep with the only instructions to remove during washing. Data was uploaded to a PC and analysed using the Actilife 6 software. Physical activity was defined with the following 3-axis vector magnitude (VM3) thresholds based on the work of Sasaki et al. [16]: sedentary 0–99 counts·min^−1^, light 100–2689 counts·min^−1^, moderate 2690–6166 counts·min^−1^, vigorous 6167–9642 counts·min^−1^ and very vigorous ≥9643 counts·min^−1^. Due to the low volume of very vigorous PA recorded this was collapsed into the vigorous category for further analysis. Daily means were calculated based on time spent at the festival for each person as follows: Wednesday 16 h (08:00–23:59); Thursday–Sunday four 24-hour periods (00:00–23:59); and Monday up to 12 h depending on when each person left the site at the end of the festival (00:00–12:00). Each participant remained on the festival site for the full duration of data collection. Total time spent in each activity level was compared to the recommended weekly moderate to vigorous physical activity (MVPA) levels set out by the National Health Service (NHS) to identify if attendance at a festival has the potential to be beneficial to health. Additionally, total step count and steps/day were calculated to indicate the level of PA each participant performed. 

Movement distances were recorded using i-gotU GT-600 GPS data loggers (MobileAction Technology Inc., New Taipei City, Taiwan) attached to the same waistbands as the Actigraph devices. Straight-line vector waypoint logging intervals were set at 60 s to allow for 160 h continuous recording, sufficient to cover the full festival period and record for the same length as the Actigraph GT3X+. Since the long recording interval could possibly result in an underestimation of total distances due to the straight-line vector measurement between acquisition points, an i-gotU GPS device was validated against a 10 Hz Catapult Minimax system (Catapult Sports, Victoria, Australia) prior to implementation in the study. Six weeks before the festival, one of the authors wore both recording devices during 10 variable distance walks (range 0.5–5 km) over a two-week period. Agreement between the devices was assessed using the Bland and Altman limits of agreement test [26] with the Catapult system assigned as the reference device due to previous validation [27]. GPS data were analysed using the proprietary Sports Analyzer software to calculate total and daily distances, mean moving speeds and movement durations. As the iGot-U data logger does not provide details of dilution of precision values to allow the accuracy of the GPS fix to be determined and used for data cleansing, individual participants’ GPS data were cleaned using a low pass filter with the upper speed limit set at 7 km·h^−1^, similar to the upper speed thresholds of 6 and 8 km·h^−1^ utilised by Kang et al. [18] and Cho et al. [28], respectively. GPS tracks were produced to identify if specific pathways were utilised during the festivals and a composite map of all participants was created in ArcGIS (Esri, Redlands, CA, USA) along with a heat map of participant density. To determine the location of different PA intensities, GPS .tcx files were correlated with the Actigraph .agd files in Actilife 6 and the resultant .kmz files were visualised in Google Earth (Google Inc., Mountain View, CA, USA).

## 3. Results

The Bland and Altman calculation displayed good agreement between the Catapult Minimax and i-gotU GT600 devices with all data points lying within ±2SD of the difference from the mean (95% CI: −162.255, 123.357) with a mean bias of −19.499. The Catapult device recorded a slightly higher mean distance of 1992 m compared to 1972 m with a difference of 0.98%. The error factor was greater at shorter distances (<1000 m differences of 9.74% and 4.24%,), while >3000 m differences of −3.87% and 0.91% were recorded.

The amount of time spent in each PA band is shown in Table 2. A mean amount of 927 min of MVPA was performed over the course of the festival (range: 473 to 1389 min). The mean daily MVPA of 155 min was higher than the recommended NHS weekly guideline of 150 min, with the highest on Saturday and Sunday and lowest on Monday. The location of PA intensities is shown in Figure 1.

GPS data loss of 171 h 22 min occurred due to battery power failure resulting in an overall data loss of 6.28%. Mean total distance was 66.08 km with the greatest individual distance of 84.51 km. The mean daily distance travelled was 11.01 km (Table 3.) and the furthest distance travelled in a single day was 20.67 km, with the greatest distances covered on the Friday, Saturday and Sunday. Mean time spent moving was 11 h 28 min 04 s per day. Distance was related to this with the greatest moving time occurring on Friday and Saturday, with the lowest moving time and lowest distance being recorded on the Monday. Daily mean moving speeds ranged from 0.66 to 1 km·h^−1^. A mean daily count of 15,661 steps was registered, with the greatest daily number of steps completed on the Sunday. A heat map (Figure 2) identifies the location of highest participant duration levels across the festival site.

## 4. Discussion

The aim of this study was to accurately determine the volume of physical activity and movement occurring at a UK outdoor festival for the first time. The purpose was to identify if the levels of moderate to vigorous physical activity through festival attendance is at the level recommended volume to aid health improvements. In the current study a mean total distance of 66.08 km and daily distance of 11.01 km were recorded, along with 927 min of MVPA during the six-day festival period. 

Mean movement speeds of 0.8 km·h^−1^ occurred over a large period of the day, with the participants producing movement for over 11 h/day. Additionally, the mean number of steps/day of 15,661 was greater than the 10,000 daily steps recommended for health benefits [29]. Movements recorded at the festival were much greater than those in previous health promotion-related movement studies. For example, in a study by Kang et al. [18] walking was measured using accelerometers and GPS. They identified that the time spent walking ranged between 27 and 48 min/day but did not measure actual distances. An intervention study on health promotion in South Africa [10] recorded movement equating to 4600 to 5400 steps/day but not distance. As this present study is the first that has attempted to measure how much festival attendees move over the course of a festival using GPS data-logging, the distances can be seen as the best estimate at present and can be used as a basis for future comparison. Whilst the combination of physical activity variables was not analysed for each attendee, examination of the wide ranges presented in Table 3 indicate that activity was highly individualised. Further analysis should focus on the classification of attendees on pre-festival activity levels, possibly using the approach developed by Bakrani et al. [30] where participants were categorised into one of four classes related to their level of active and non-active physical activity. These approaches may identify if the higher levels of physical activity occurring at the festival arise because the participant has previously been or is naturally more likely to be active, or if the increase in PA occurs in those who previously were inactive. 

Mean MVPA performed over the festival period of 927 min was greater than the NHS weekly guidelines [5] of 150 min. Even the daily amount of 155 min was greater than the weekly recommendation. As such, physical activity at the festival was above the required level to elicit health benefits [14] and could be included as an activity to promote healthy behaviour. The advantage over common activities that are usually recommended is that PA at festivals occurs solely through attendance and not as a structured activity, where participants do not have to specifically go to a class or gym to achieve a large volume of MVPA. This can be fitted to the Social–Ecological Model [31] for promoting physical activity, where human behaviour towards PA is affected by social and physical environmental factors. In this case, PA is performed in a natural environment that creates a positive influence on the festival attendee, aligned to a human–environment interaction that is flexible and adaptable [32] but related to the festival schedule and site. Music festivals are unique experiences that combine increases in general movement, e.g., walking, with pleasurable activates that produce more vigorous movement, e.g., dancing, that mask the physical and health benefit. This produces an unintended health benefit through the increase in PA levels to and above the recommended daily values. Whilst this festival attendance only represented six days activity duration, people often attend more than one festival in a year and high levels of physical activity may be repeated throughout the summer. With high amounts of MVPA evident at this festival and the possibility of further accumulation of MVPA it is proposed that attendance at festivals could be included as a modality of attaining the levels of PA beneficial to health alongside the commonly prescribed fitness activities to reach the weekly goal. Further research is necessary to understand if similar physical activity levels occur at other festivals or if the increase in physical activity observed during the festival is continued post-festival, rather than a boom–bust scenario where physical activity drops off after the festival has ended.

Previous studies have included definitions of sedentary and active behaviours [33]. Whilst these are not categorised in the present study, knowledge of the festival schedule can be used to identify specific activities. These include walking to and from the car on the Wednesday and Monday. This would normally fall as light PA but through the addition of kit transportation this movement becomes high intensity PA. General movement around the site can be classified as standing or walking based on the mean moving speeds recorded by the GPS device of between 0.66 and 1 km·h^−1^. However, moderate and vigorous intensity activity is likely to occur when watching and dancing to musical acts, as shown through the trend of increased MVPA on the Friday–Sunday when the majority of entertainment happened. As seen in Figure 1 it is possible to identify the location of moderate and vigorous PA, for example with MVPA occurring at the Pyramid Stage, Dance Village and SE Corner. Previous research has reported the benefits of dance on physical health [34], with improvements in body composition, cardiovascular and musculoskeletal function all evident, alongside psychosocial benefits [35]. In the ‘Dance for Health’ program, Schroeder et al. [36] reported that weekly sessions of dance elicited target heart rates and perceived exertion levels commensurate with moderate physical exertion. Additionally, enjoyment was rated highly by all participants, indicating the benefits of dance as a form of physical activity that may overcome the perceived negative aspects of traditional exercise. Therefore, this strongly suggests that the combination of movement and ‘dance’ experienced by attendees at a music festival will further aid achievement of recommended physical activity and health benefits.

An additional benefit of using GPS data-loggers is the ability to combine the data with geographic information systems (GIS) [20] to identify movement in relation to the surrounding environment [21]. This could inform festival organisers of such information as areas of high traffic. Figure 2 identifies the areas of high and low participant duration. Using this, it is possible to identify the camping locations and distinct pathways around the site and location of stages and fields visited. Glastonbury Festival is unique for a large-scale festival in that once inside a perimeter fence, attendees have free access with stages and camping areas mixed together and no travel to external campsites necessary, so distances and activities recorded were those solely conducted at the festival. A similar system was utilised by Paz-Soldan et al. [37] to capture human mobility in Iquitos, Peru to highlight disease infection patterns. It is interesting to note the GPS track patterns indicate the most popular stages and venues on site and path selection is the utilisation of the most efficient routes between these, and this can be attributed to knowledge of the site layout. Additionally, this information has health and safety implications as it is possible to understand large-scale crowd movement [38]. 

Data loss is a methodological problem in GPS-based studies with levels reported between 2.5 and 92% [18,19]. Reasons for data loss include power loss, poor signal and environmental interference [18,19,20]. Data loss in the current study was 6.28%, primarily due to the loss occurred during data cleaning where erroneous GPS values were manually removed. Additional data loss was due to power loss on the Monday in two devices and an auto-powering off in a third device. The fact that the festival is located in a rural environment with a very small number of buildings reduces the possibility of signal interference but a high density of festival attendees, especially at the main stages, may result in GPS drift as the device is blocked by other people due to being worn on the waist. Any such data loss can compromise the strength of a GPS study [19] and caution should be taken when viewing the GPS data. However, the distance walked as measured by the GPS data logger is similar to that calculated from the Actigraph step count, assuming an average step length of 0.7 m, with a total distance of 65.76 km and a daily mean distance of 10.96 km. The sampling approach utilized in the current study may have influenced the distances and physical activity levels, as the convenience sample may have selected more physically active festival attendees, and this may have skewed the recorded data upwards. Additionally, the inclusion of regular festival attendees to standardise data collection and improve adherence to the study renders an unbiased recruitment procedure inappropriate. However, the characteristics of the participants include a large age range and pre-festival activity levels and can be seen as representative of a general festival population. Length of the festival, lack of power availability and connection with the outside world for six days necessitated the use of a GPS device at a low sample frequency to ensure data collection for the full festival period. A comparison with a previously validated GPS device, recording at 10 Hz, was conducted prior to the festival. A Bland and Altman [26] comparison found that the two devices were similar with all data points lying within the 95% confidence limits of agreement. A small bias of −19.499 and difference of −0.98% in absolute distances were present. This may have led to an over-estimation in distances, which could have been further compounded by the larger distances at the festival than the range used in the validation. However, actual replication in validating journeys in excess of 10 km would have been impractical. No issues were found with the Actigraph device as it was previously validated [15,17] and the longer recording period in the current study did not cause any acquisition errors. Finally, it is possible that participants altered their movement and physical activity as a result of knowingly wearing the tracking devices, producing greater measured values than would have naturally occurred. In an attempt to minimize this risk, all participants were familiarised with the devices before the festival and advised to wear them as part of their normal clothing to make them less obvious throughout each day of the festival. It is assumed that any potential increase will have been negligible compared to the actual increases in PA due to festival attendance.

## 5. Conclusions

This study provided a unique opportunity to record the volume of PA that occurred at an outdoor UK festival using objective measures of GPS data and accelerometer-based movement analysis. Large distances of between 47 and 85 km with a mean of 66.08 km and MVPA six times greater than the recommended guidelines for health benefits were found. It can be suggested that attending large-scale outdoor UK festivals offers the possibility to be used as an additional method for attaining physical activity guidelines alongside commonly suggested fitness activities. It is obvious what we need in the UK to get the population walking and jumping around—a weekly Glastonbury Festival!

## Figures and Tables

**Figure 1 ijerph-16-02105-f001:**
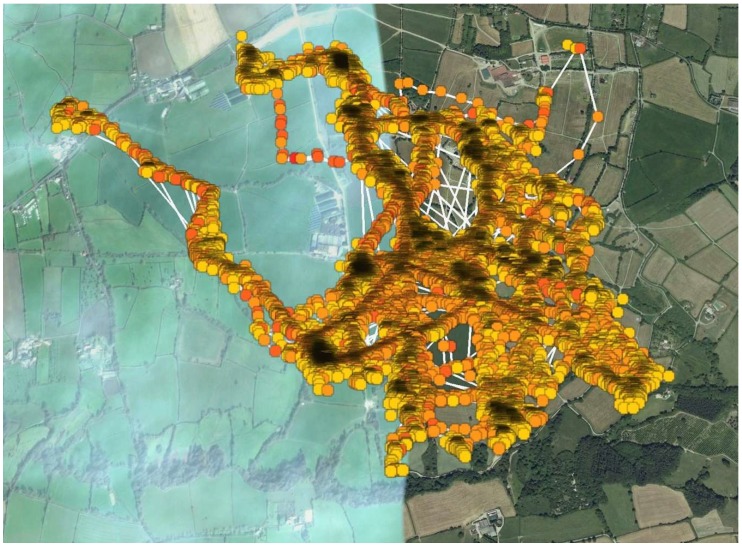
Location of physical activity intensities for all participants around the festival site. Circle colour indication: Yellow = Light; Orange = Moderate; Red = Vigorous. (Dark areas indicate increased density of physical activity; white lines represent additional movement pathways with minimal physical activity registered).

**Figure 2 ijerph-16-02105-f002:**
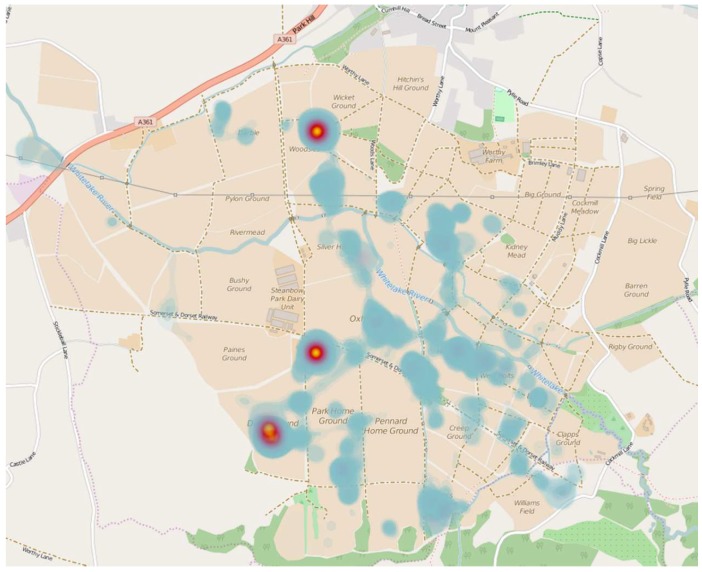
Heatmap indicating areas of highest participant duration during the festival (High—Yellow; Medium—Blue).

**Table 1 ijerph-16-02105-t001:** Participant characteristics.

Characteristic	Mean (±SD); Range
Age (years)	32 ± 8; 22–57
Height (cm)	179.09 ± 4.92; 169–186
Weight (kg)	81.32 ± 21.51; 58–152
Attendance (number of times)	5 ± 2; 1–10

**Table 2 ijerph-16-02105-t002:** Time spent (min) in physical activity threshold bands (Mean value ± SD; Range).

	Sedentary	Light	Moderate	Vigorous	Total Time in MVPA *
**Wednesday**	281 ± 198; 38–735	565 ± 180; 263–919	118 ± 70; 2–232	16 ± 15; 0–48	134 ± 76; 3–262
**Thursday**	708 ± 205; 394–1093	564 ± 197; 202–935	143 ± 43; 86–239	25 ± 24; 1–94	168 ± 50; 87–246
**Friday**	746 ± 177; 485–1036	531 ± 177; 227–821	142 ± 44; 65–221	22 ± 33; 0–141	163 ± 53; 67–272
**Saturday**	817 ± 216; 431–1372	420 ± 214; 44–863	183 ± 61; 24–279	20 ± 23; 0–79	203 ± 72; 24–344
**Sunday**	729 ± 190; 376–1058	519 ± 201; 238–900	173 ± 64; 68–291	19 ± 18; 0–53	192 ± 75;77–344
**Monday**	393 ± 194; 87–810	398 ± 171; 151–701	60 ± 31; 3–124	8 ± 9; 1–24	68 ± 32; 3–126
**Total**	3673 ± 789; 2287–5369	2997 ± 761; 1619–4716	819 ± 208; 468–1190	109 ± 85; 5–262	927 ± 252; 473–1389
**Mean**	612 ± 132; 381–895	500 ± 127; 270–786	136 ± 35; 78–198	18 ± 14; 1–44	155 ± 42; 79–231

* Moderate to vigorous physical activity (sedentary 0–99 counts·min^−1^, light 100–2689 counts·min^−1^, moderate 2690–6166 counts·min^−1^, vigorous 6167–9642 counts·min^−1^ and very vigorous ≥9643 counts·min^−1^) Sasaki et al. [16].

**Table 3 ijerph-16-02105-t003:** Movement data during the festival (Mean ± SD; Range).

	Distance (km)	Movement Time (hr:mm:ss)	Average Moving Speed (km/h)	Step Count
**Wednesday**	9.25 ± 4.66; 0.52–15.90	7:04:00 ± 2:24:23; 2:01:32–11:31:51	1.00 ± 0.49; 0.10–1.90	14,427 ± 5185; 3511–25,398
**Thursday**	12.63 ± 2.63; 8.75–17.38	13:22:30 ± 3:21:20; 2:38:22–16:56:06	0.84 ± 0.50; 0.40–2.70	16,406 ± 4269; 11,086–24,145
**Friday**	11.65 ± 2.49; 7.02–17.58	14:44:00 ± 2:56:08; 10:25:53–20:03:45	0.66 ± 0.24; 0.30–1.35	16,030 ± 3116; 11,445–21,574
**Saturday**	13.82 ± 3.65; 2.61–19.95	14:50:15 ± 3:06:55; 9:35:12–20:02:35	0.74 ± 0.28; 0.20–1.30	19,682 ± 5809; 2365–30,832
**Sunday**	13.50 ± 4.65; 5.07–20.67	13:22:13 ± 3:15:44; 5:33:07–17:35:32	0.86 ± 0.57; 0.30–2.85	20,013 ± 5938; 8793–28,989
**Monday**	5.22 ± 2.28; 2.16–11.18	5:25:24 ± 1:45:44; 2:22:14–8:29:21	0.72 ± 0.30; 0.20–1.40	7407 ± 3735; 1630–15,540
**Total**	66.08 ± 11.34; 47.03–84.51	68:48:22 ± 10:44:44; 40:59:44–86:38:05		93,966 ± 18,284; 73,543–135,817
**Daily Mean**	11.01 ± 1.89; 7.84–14.09	11:28:04 ± 1:47:27; 6:49:57–14:26:21	0.80 ± 0.28; 0.40–1.63	15,661 ± 3047; 11,257–22,636

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
