# Peer review of "Putting the Boom, Boom, Boom into Physical Activity and Health: Music Festivals as a Positive Health Alternative to Couch Fandom"

_ijerph, 2019, doi:10.3390/ijerph16122105_

Round 1

Reviewer 1 Report

This paper is a worthy antidote to the overwhelming number of studies that associate physical activity with a narrow range of 'sports'. Physical activity promotion is hindered/constrained as a result of this lack of imagination so it is heartening to see this original and innovate paper offer us an alternative perspective. 

The paper is well organised, detailed and concisely written. It outlines a research intervention at the Glastonbury Festival, and explains the research method clearly and with appropriate substantiation before analysing the data and drawing conclusions that question a number of everyday assumptions about (a) the experience of Festival goers and (b) settings where physical activity is prevalent. 

I have some minor comments that can be easily addressed, but my three more substantive concerns are as follows:

There is a slight danger that the paper will overstate the obvious in the sense that walking will be necessary in many contexts where key aspects of an experience are located in different spaces. So any leisure experience where walking is a component (visiting stately homes; rambling; going to LegoLand), offers scope for fulfilling PA targets. I think it would be sensible to make clear what is unique about music festivals in this regard - is it the combination of different activities? The broader emphasis on escape and lifestyle? 

If music and dance are central to the argument, I think that reference should be made to the body of knowledge that assesses the relationship between dance and physical activity. This element of the argument could be woven in further, by commenting more explicitly (in the Results and Discussion) about the moments/episodes where vigorous activity took place. 

There is a need to acknowledge the small possibility that PA levels may have been affected by the subjects' awareness of their role and of the purpose of the research. 

Each of these issues can be addressed very simply so I do not think that they count as major revisions but I do think that they should be addressed. 

In terms of minor amendments, I draw attention to:

22-24 grammar/clarity

37 - not sure why 'expensive' is a necessary adjective here

42-43 grammar

66-69 is 'identify' the right verb here, perhaps 'explore the extent to which...' - something more open perhaps?

A further spell/grammar check would be sensible. 

Author Response

Responses to the reviewer are included in the attached document

Reviewer 2 Report

This is a novel and interesting study, providing data on the volume and duration of physical activity accumulated during a large and popular music festival in the UK.

The findings are certainly noteworthy and the fact that the average daily number of steps exceeds the 10,000 steps recommendation is particularly interesting. However, I think the authors may be  overly ambitious in their aim of considering music festival attendance as an added method for attaining physical activity guidelines. While the data are solid, the link to physical activity recommendations is complex, with a need for more data (i.e., future studies) and more comprehensive considerations.

My recommendation is to change the second aim of the study (lines 66-68) to the one mentioned in line 101 (i.e., comparison with recommended MVPA levels “to identify if attendance at a festival has the potential to be beneficial to health”).

Further, I would like to see some more considered arguments in the discussion regarding the potential role of music festival attendance in relation to physical activity guidelines. There is some attention to the limitations of the study (though not very explicitly) in relation to future research (lines 180-186, and the argument in lines 183-186 is very important) but this could be extended.  

Some things to consider, for example:

E.g., future studies should monitor/assess physical activity levels at other times as well as pre- and post- festival. There could, for example, well be a compensation effect of low physical activity after the festival. Would this cancel out the levels achieved during the festival?

People are unlikely to go to music festivals for the purpose of getting fitter. However, for those whose with an interest in music/dance/outdoor festivals, the environmental characteristics of the festival can act as environmental ‘nudges’ to get people to walk/move more. To what extent this encourages people who would not otherwise meet physical activity guidelines to do so remains to be investigated.

Furthermore, we don’t know the characteristics (e.g., demographics, health status, physical activity levels) of people who typically attend such festivals. Is this likely to be a special group?

Another issue to address in the discussion is whether there could have been a reactivity effect: could participants have increased their walking because they were aware of the purpose of the study?

More specific comments and queries are provided below by line.

Lines 35-58. For readers not familiar with the specific of activity monitors, it would be useful to give a brief explanation of what sort of activity monitors exist and what they measure specifically, before discussing studies that use them. E.g., in line 46, you mention that these devices record “counts of movement”; could you explain this is more detail, is it just steps or also other types of movement?

Lines 64-65: Could you provide a source for these anecdotal data, even if they are only known to the authors? Could these perhaps be referred to as “informal exploratory work”?

Lines 70-73: Please provide more detail about the participant recruitment. Was there any attempt at random or purposeful selection (e.g., based on range of physical activity levels), or was it predominantly a convenience sample?  If the latter, address this as a limitation in your discussion.

Why was there a need to include “regular festival attendees”? Please provide more details for this argument (also related to argument in lines 234-237:).

Lines 71-73: How was pre-festival physical activity level assessed? Furthermore, could you add the numbers to indicate how many participants had low, medium, and high levels respectively?

Lines 86-94: related to my earlier comment, please add a brief explanation [this could be done in the earlier brief introduction] of what exactly the Actigraph measures in terms of physical activity. I assume the ‘counts per minute’ refers to number of steps taken, however, you refer to “physical activity levels were recorded” (line 86), so to readers not familiar with the recording device, this link is not immediately evident.

Lines 110-123: I found the description in this section somewhat confusing. My understanding is that there was a validation procedure to assess agreement between the i-gotU GPS and the Catapult Minimax system, which involved someone (who was this and was this done by only one person?) wearing both devices for two weeks. However, I was struggling to distinguish whether the remainder of the paragraph refers to this validation procedure or to the data analysis of the current study. It might be clearer if you add the word “Because” in line 107 “Because the long recording interval…”, and continue the sentence in line 108 rather than start a new one: “…acquisition points, an i-gotU GPS device as validated…”

Line 104. What was the spread of data loss across the participants? How was the data loss taken into account in the calculation of means, etc?

Lines 137-183: Figure 1: what is the meaning of the dark spots and lines, and of the white lines?

Line 148 and Figure 2: What exactly does ‘heat’ mean here? There is an issue here about relevance to the purpose of the study. Although interesting that the GPS data allow for an indication of highest participant duration during the festival, I’m not sure what these data add to the purpose of the study as focused on the volume and intensity of physical activity levels during the festival.

Further to this, I wonder about the relevance of the paragraph between lines 187-200 in the discussion, given the the purpose of the study. I think you could leave this paragraph out (as well as figure 2). Alternatively, provide a good argument for including this information and add it to your aim.

Lines 163-166: Again you mention anecdotal accounts, but without more detail of what these consisted of, this information is less meaningful. Consider either adding more detail on what was found in these early explorations, or leaving out this argument.

Lines 167-169: This sentence is incomplete.

Lines 170-174: What is the main argument here? Is it the fact that the studies mentioned recorded walking minutes or steps per day but not distances, or is it that the amount of walking was less than in the current study? In the latter case, please provide more detail about the context of the other studies, and also consider the participants. Walking in everyday life, and interventions aimed at enhancing physical activity levels in insufficiently active populations provide very different conditions than a very specific event or physical activity levels measured in people who were already reasonably active.

Line 177-178: this is a bit of a convoluted sentence. What exactly do you mean by ‘valid’?

Minor editorial comments:

Line 70:  Delete ‘and’ and start a new sentence with “Characteristics”

Line 181: perhaps replace “or based on” by “possibly using”

Line 216: change ‘become’ to ‘becomes’

Author Response

(The authors gave the same response as above.)

Reviewer 3 Report

This is a really interesting study. I can’t comment directly on the methods utilized for collecting data but the use of the accelerometer and GPS tracker together appears to be the best option for this level of data recording. I think it is important to start thinking outside of the box when its comes to exercise prescription however, given that this is not something that would be done on a regular basis perhaps you could relate some of the findings to other studies that have been done to try and increase physical activity outside of the gym environment.

Consider rewording ‘sports participation of 25.2% partaking in zero sessions.’ Line 33

Line 201 on. I find this paragraph to be the most informative from the standpoint of the understanding of the behavioral implications of having individuals meet PA guidelines without having to think about this engagement. I wonder if you could tie in the social ecological model as commonly used to focus on the need for reducing sedentary behavior and increasing movement time when it comes to meeting the PA guidelines. This has been the model that has been proposed for understanding the behavioral challenges of meeting the PA guidelines. (Specifically line 207).

Author Response

(The authors gave the same response as above.)

Round 2

Reviewer 2 Report

I would like to thank the author for the detailed responses to my comments and for the changes made. I am happy with the way you have addressed my comments.